# Endoplasmic Reticulum Stress Mediated NLRP3 Inflammasome Activation and Pyroptosis in THP-1 Macrophages Infected with *Bacillus Calmette-Guérin*

**DOI:** 10.3390/ijms241411692

**Published:** 2023-07-20

**Authors:** Xueyi Nie, Boli Ma, Lei Liu, Xiaotan Yuan, Mengyuan Li, Yueyang Liu, Yuxin Hou, Yi Yang, Jinrui Xu, Yujiong Wang

**Affiliations:** 1School of Life Sciences, Ningxia University, Yinchuan 750021, China; 18794898774@163.com (X.N.); 15695013087@163.com (B.M.); liulei7@vip.163.com (L.L.); 18393712338@163.com (X.Y.); 20180044@nxmu.edu.cn (M.L.); 18395273708@163.com (Y.L.); 18209689779@163.com (Y.H.); yangyi@nxu.edu.cn (Y.Y.); 2Key Laboratory of Ministry of Education for Conservation and Utilization of Special Biological Resources in the Western, Ningxia University, Yinchuan 750021, China

**Keywords:** endoplasmic reticulum stress, NLRP3 inflammasome, pyroptosis, *Bacillus Calmette-Guerin*, THP-1 macrophages

## Abstract

Tuberculosis (TB) is a zoonotic infectious disease caused by *Mycobacterium tuberculosis* (Mtb). Mtb is a typical intracellular parasite, and macrophages are its main host cells. NLRP3 inflammasome-mediated pyroptosis is a form of programmed cell death implicated in the clearance of pathogenic infections. The bidirectional regulatory effect of endoplasmic reticulum stress (ERS) plays a crucial role in determining cell survival and death. Whether ERS is involved in macrophage pyroptosis with Mtb infection remains unclear. This article aims to explore the regulation of the NLRP3 inflammasome and pyroptosis by ERS in THP-1 macrophages infected with *Mycobacterium bovis Bacillus Calmette-Guérin* (BCG). The results showed that BCG infection induced THP-1 macrophage ERS, NLRP3 inflammasome activation and pyroptosis, which was inhibited by ERS inhibitor TUDCA. NLRP3 inhibitor MCC950 inhibited THP-1 macrophage NLRP3 inflammasome activation and pyroptosis caused by BCG infection. Compared with specific Caspase-1 inhibitor VX-765, pan-Caspase inhibitor Z-VAD-FMK showed a more significant inhibitory effect on BCG infection-induced pyroptosis of THP-1 macrophages. Taken together, this study demonstrates that ERS mediated NLRP3 inflammasome activation and pyroptosis after BCG infection of THP-1 macrophages, and that BCG infection of THP-1 macrophages induces pyroptosis through canonical and noncanonical pathways.

## 1. Introduction

Tuberculosis (TB) is a chronic infectious disease caused by *Mycobacterium tuberculosis* (Mtb) infection, and Mtb mainly proliferates within the alveolar macrophages of the infected host [1]. Macrophages, which are innate immune cells, play a critical role in protecting the host from a variety of bacterial and other pathogens. Once Mtb enters macrophages, it resides within the phagosome and subsequently disrupts its normal function. Macrophages play a crucial role in eliminating Mtb within the phagosome by producing reactive oxygen and nitrogen. They also initiate a proinflammatory response that recruits immune cells to the infection site, resulting in the formation of granulomas. These granulomas effectively prevent bacterial replication and dissemination [2]. In addition, Mtb can also evade host cell clearance by preventing the maturation process of macrophages [3]. During these processes, Mtb, *Mycobacterium bovis* (*M. bovis*) and *Mycobacterium bovis Bacillus Calmette-Guérin* (BCG) have shown variations in virulence, host range and metabolism. BCG is the only vaccine available for human use against Mtb [4], studying the immunological effects of BCG’s interaction with infected host cells can enhance our understanding of tuberculosis pathogenesis.

The endoplasmic reticulum (ER) is an organelle that plays a crucial role in maintaining homeostasis in eukaryotic cells. Various pathogens or adverse stimuli can induce endoplasmic reticulum stress (ERS) [5,6,7]. ERS leads to the accumulation of unfolded or misfolded proteins in the ER, impairing its normal physiological functions. In mild ERS, ER protects cells from damage by activating unfolded protein response (UPR) to remove misfolded proteins, and in excessive or persistent ERS, UPR stimulates autophagy, apoptosis and pyroptosis [8]. Downstream of UPR, there are three receptors: inositol-requiring enzyme 1 (IRE1), double-stranded RNA-activated protein kinase-like endoplasmic reticulum kinase (PERK) and activating transcription factor 6 (ATF6) [9,10,11]. Studies have shown that Mtb can induce ERS in infected macrophages, thereby changing the fate of cells and Mtb [7,12].

The NLRP3 inflammasome is a crucial pattern recognition receptor of the innate immune system, playing a vital role in protecting the host against bacterial, fungal and viral infections [13,14]. NLRP3 inflammasome activation generally requires two steps: initiation (signal 1) and activation (signal 2). The initiation process is triggered by pattern recognition receptor signaling, such as Toll-like receptor (TLR) 4 or tumor necrosis factor (TNF) signaling, whose subsequent maturation of pro-IL-1β and pro-IL-18 into IL-1β and IL-18 and release are promoted through a nuclear factor-κB (NF-κB)-dependent pathway. The activation process (signal 2) is induced by various pathogen-associated molecular patterns (PAMPs) and damage-associated molecular patterns (DAMPs), including extracellular ATP, pore-forming toxins, RNA viruses and particulate matter. Mitochondrial dysfunction, reactive oxygen species (ROS) generation and lysosomal damage are all involved in the activation of NLRP3 inflammasome assembly [15]. It has been shown that ERS triggers the UPR, which in turn activates the NLRP3 inflammasome [16].

Pyroptosis is a form of inflammatory cell death that can be triggered by either a canonical Caspase1-mediated pathway or a noncanonical Caspase4/5/11-mediated pathway [17]. In both of these pathways, the corresponding caspases have the ability to cleave gasdermin-D (GSDMD) to produce GSDMD-N that can form pores via oligomerization at the plasma membrane [18]. This process involves the expulsion of cellular contents and the release of IL-1β and IL-18 [19,20]. Moreover, Caspase3/8 can induce pyroptosis: Caspase3 cleaves GSDME to generate the pyroptosis mediator GSDME-N, while Caspase8 cleaves GSDMD in response to TAK1 inhibition [21,22]. Various factors, such as intracellular bacterial infection, have been shown to stimulate pyroptosis. As a result, immune cells are recruited to the site of infection to clear the bacteria and fight against further infection [23].

This study investigated whether ERS is involved in NLRP3 inflammasome activation and pyroptosis after infection of THP-1 macrophages with BCG. Our results showed that BCG-infected THP-1 macrophages induce ERS, activate the NLRP3 inflammasome and trigger pyroptosis, and ERS is involved in NLRP3 inflammasome activation and mediated pyroptosis. These results provide a new molecular mechanism for BCG-induced macrophage pyroptosis, and provide a new basis for further research on the pathogenesis of TB and its treatment.

## 2. Results

### 2.1. BCG Infection Induced THP-1 Macrophage ERS, NLRP3 Inflammasome Activations and Pyroptosis

To investigate the effects of BCG infection on THP-1 macrophage ERS, NLRP3 inflammasome and pyroptosis, THP-1 macrophages were infected with BCG for 2 h, 6 h, 12 h, 24 h and 48 h. Western blotting showed that BCG infection increased the expression of ERS-related proteins (IRE1α, PERK, ATF6, GRP78 and CHOP [24]), NLRP3 inflammasome-related proteins (NLRP3, ASC and Pro-Caspase1) and pyroptosis-related proteins (GSDMD-N, IL-1β, Cleaved-IL-1β and IL-18) of THP-1 macrophages in a time-dependent manner (Figure 1A–C). TEM observation of cell morphology indicated that the cell morphology and structure in the Ctrl group were normal, the organelles in the cytoplasm were clear, and the cell membrane was intact. In the BCG group, the cell membrane was ruptured, the cytoplasmic ribosomes were significantly lost, most of the mitochondria had swollen (the cristae were broken and dissolved, the matrix was lost, they were in a flocculent structure and the mitochondria were vacuolated), and the rough endoplasmic reticulum was also expanded. (Figure 1F). CCK-8 (Figure 1D) and LDH (Figure 1E) showed that cell viability decreased with the prolongation of BCG infection time. These findings suggested that BCG infection could induce THP-1 macrophage ERS, NLRP3 inflammasome activation and pyroptosis.

### 2.2. TUDCA Inhibits BCG-Infected THP-1 Macrophage NLRP3 Inflammasome Activity and Pyroptosis

To explore the regulatory effects of ERS on NLRP3 inflammasome activity and pyroptosis induced by BCG infection, THP-1 macrophages were pretreated with ERS inhibitor TUDCA, and then infected by BCG. Western blotting showed that the protein expression of IRE1α, PERK and ATF6 in THP-1 macrophages was significantly increased with BCG infection (Figure 2A), which was inhibited by TUDCA. The expression of NLRP3, Pro-Caspase1 and ASC proteins showed the same trend (Figure 2B). Immunofluorescence showed that the expression of NLRP3 of THP-1 macrophages was increased with BCG infection, and was alleviated by TUDCA (Figure 2C). These results indicated that ERS mediated NLRP3 inflammasome activation in THP-1 macrophages upon BCG infection.

Similarly, the upregulated expression of pyroptosis-associated proteins Cleaved Caspase1, GSDMD-N, IL-1β, Cleaved-IL-1β and IL-18 in THP-1 macrophages caused by BCG infection was inhibited by TUDCA (Figure 3A). TEM observation of cell morphology showed that the cell morphology and structure of the control group were normal, with intact cell membranes and irregular nucleus. In the BCG group, clear signs of cell pyroptosis were observed, including cell membrane rupture, significant loss of cytoplasmic ribosomes, mitochondrial swelling and rough endoplasmic reticulum expansion. After TUDCA pretreatment, there were a few discontinuous areas in the cell membrane and mild swelling of the mitochondria, which attenuated the symptoms caused by BCG infection-induced pyroptosis (Figure 3B). ELISA showed that BCG infection raised the extracellular release of IL-1β and IL-18 (Figure 3C,D), while TUDCA pretreatment reduced the release of these inflammatory cytokines. Cell viability was assessed via CCK-8 (Figure 3E) and LDH (Figure 3F) assays showed that the cell viability decreased and the activity of LDH increased significantly in the BCG-infected group compared with the control group, and TUDCA pretreatment increased the cell viability and reduced the release of LDH. Together, these data demonstrated that ERS mediated pyroptosis induced by BCG infection.

### 2.3. MCC950 Inhibits BCG-Infected THP-1 Macrophage NLRP3 Inflammasome Activation and Pyroptosis

To verify the regulatory role of the NLRP3 inflammasome on pyroptosis during BCG infection, THP-1 macrophages were pretreated with NLRP3 inhibitor MCC950 and then infected by BCG. Western blotting showed that the expressions of NLRP3, Pro-Caspase1 and ASC in THP-1 macrophages were significantly upregulated with BCG infection (Figure 4A–D), which was inhibited by MCC950. Immunofluorescence analysis revealed that the expression of NLRP3 in THP-1 macrophages was upregulated upon BCG infection. However, this upregulation was mitigated by MCC950 treatment (Figure 4E). The upregulated expression of Cleaved Caspase1, GSDMD-N, IL-1β, Cleaved-IL-1β and IL-18 in THP-1 macrophages caused by BCG infection was also inhibited by MCC950 (Figure 5A–F). TEM observation of cell morphology indicated that the cell morphology and structure of the control group were normal, with intact cell membranes and clear structure of each organelle in the cytoplasm. The BCG group exhibited evident characteristics of pyroptosis, including cell membrane rupture, rough endoplasmic reticulum expansion, cytoplasmic ribosome loss and mitochondrial swelling. Pretreatment with MCC950 resulted in a slight loss of cytoplasmic ribosomes and mild mitochondrial swelling, which alleviated the pyroptosis induced by BCG infection (Figure 5G). ELISA showed that BCG infection raised the extracellular release of IL-1β and IL-18 (Figure 5H,I), while MCC950 pretreatment decreased these inflammatory cytokines. Cell viability was assessed via CCK-8 (Figure 5J) and LDH (Figure 5K) assays showed that compared with the control group, the cell viability decreased and the activity of LDH increased significantly in the BCG-infected group, while MCC950 pretreatment increased the cell viability and reduced the release of LDH. Together, these data suggest that NLRP3 inflammasome activation mediated pyroptosis induced by BCG infection.

### 2.4. BCG-Infected THP-1 Macrophage Trigger Pyroptosis via Canonical and Noncanonical Pathways

To clarify the pathway of pyroptosis in THP-1 macrophages infected with BCG, THP-1 macrophages were pretreated with the Caspase1-specific inhibitor VX-765 and the pan-Caspase inhibitor Z-VAD-FMK prior to BCG infection. Western blotting showed that GSDMD-N, IL-1β, Cleaved-IL-1β and IL-18 expression were significantly upregulated upon BCG infection, and pretreatment THP-1 macrophages with VX-765 and Z-VAD-FMK prior to BCG infection effectively suppressed pyroptosis, with Z-VAD-FMK exhibiting a greater inhibitory effect (Figure 6A–E). ELISA showed that the concentration of IL-1β and IL-18 in the cell culture supernatant was consistent with the previously mentioned findings (Figure 6F,G), and BCG-induced increases in LDH released from THP-1 macrophages were reduced in the context of such pretreatment (Figure 6H). These results suggested that BCG induces pyroptosis in THP-1 cells through canonical and noncanonical pathways.

## 3. Discussion

Tuberculosis remains a significant global public health concern. The Mtb responsible for TB initially infects the distal airways and then spreads to the interstitium of the lung. Macrophages are the primary target cells for Mtb, as they possess the ability to phagocytose these bacteria. However, Mtb can thrive and multiply in macrophages due to its ability to interfere with phagolysosome formation [25,26]. Thus, the form of death of infected macrophages plays a pivotal role in the outcome of Mtb infection. Gaining a better understanding of the process of macrophage death triggered by Mtb infection will help in identifying new targets for host-directed therapy (HDT) of tuberculosis.

Pyroptosis is a form of cell death that is mediated by the GSDM family of proteins, including GSDMD and GSDME. Studies have shown that Mtb infection can trigger macrophage pyroptosis, leading to the release of proinflammatory factors and exacerbating tissue damage [27]. Our current study confirmed this finding: this study found that BCG infection caused the upregulation of pyroptosis-related molecules GSDMD-N, Cleaved Caspase1, IL-1β, Cleaved-IL-1β and IL-18 in macrophages, the cell membrane rupture, the loss of cytoplasmic ribosomes and mitochondrial swelling, supporting the ability of BCG to induce macrophage pyroptosis.

Previous studies demonstrated that various diseases, including I/R injury [28] and Periodontitis [29], are associated with both canonical and noncanonical pathways of pyroptosis. Therefore, we explored the pathway of pyroptosis after BCG-infected THP-1 macrophages. In this study, we pretreated cells with the VX-765 and Z-VAD-FMK, to investigate their effects on pyroptosis. Our results showed that both inhibitors effectively inhibited pyroptosis. However, Z-VAD-FMK exhibited a greater downregulation of pyroptosis-related proteins compared to VX-765. These findings suggest that the pathways of pyroptosis induced by BCG infection of THP-1 macrophages involve both canonical and noncanonical pathways.

Pyroptosis is a unique form of cell death that involves the activation of the NLRP3 inflammasome [30]. The NLRP3 inflammasome consists of the NLRP3 receptor, the ASC adaptor protein and the caspase-1 effector protein, which can be activated by a range of stimuli. Studies have shown that NLRP3 inflammasome/pyroptosis is related to various diseases, including diabetic retinopathy [31] and Parkinson’s disease [32]. However, the relationship between the NLRP3 inflammasome and pyroptosis in BCG-infected THP-1 macrophages remains unclear. This study provided evidence that BCG infection can activate the NLRP3 inflammasome in THP-1 macrophages by upregulating NLRP3 inflammasome-related molecules NLRP3, Pro-Caspase1 and ASC. Additionally, the results suggest that NLRP3 inhibitor MCC950 may inhibit pyroptosis following BCG infection, which is consistent with past findings [13].

The ER plays a vital role in eukaryotic cells, being responsible for secretory protein synthesis, folding and modification, regulation of lipid synthesis, maintenance of intracellular calcium ion homeostasis, etc. [33,34] Both pathological and physiological conditions can contribute to protein misfolding and the consequent induction of ERS, which has been found to be significant in the development of tuberculosis [12,34]. Previous studies have demonstrated that ERS is involved in the activation of the NLRP3 inflammasome and the induction of pyroptosis in macrophages infected with Mtb and *M. bovis* [34,35,36]. However, the relationship between ERS, the NLRP3 inflammasome and pyroptosis after BCG-infected macrophages remains unclear. Therefore, we investigated the potential involvement of ERS in the activation of NLRP3 inflammasome and pyroptosis in macrophages infected with BCG. This study has confirmed that BCG infection can cause ERS by increasing ERS-related proteins IRE1α, PERK and ATF6 in THP-1 macrophages. Additionally, ERS inhibitor TUDCA infection can prevent NLRP3 inflammasome activation and pyroptosis in BCG-infected THP-1 macrophages. These results suggest that ERS mediates NLRP3 inflammasome activation and pyroptosis in THP-1 macrophages infected with BCG.

This study employed utilized corresponding inhibitors to investigate the correlation between ERS, the NLRP3 inflammasome and pyroptosis in THP-1 macrophages infected with BCG. Ultimately, these data demonstrated BCG infection to induce ERS, which in turn mediated NLRP3 inflammasome activation and pyroptosis. Additionally, the study revealed that BCG infection of THP-1 cells can trigger pyroptosis through both canonical and noncanonical pathways.

## 4. Materials and Methods

### 4.1. Antibodies and Reagents

Key reagents used included Fetal Bovine Serum (FBS, 10099141C, Gibco, Carlsbad, USA); RPMI-1640 (C11875500BT, Gibco, Carlsbad, CA, USA); β-mercaptoethanol (M8211, Solarbio, Beijing, China); PMA (P1585, Sigma-Aldrich, St. Louis, MO, USA); Middlebrook 7H9 Broth (M1315, BD, San Jose, CA, USA); Middlebrook ADC Enrichment (211887, BD, San Jose, CA, USA); M-PER (78501, Thermo Fisher Scientific, Waltham, MA, USA); Protease inhibitor Cocktail (78442, Sigma-Aldrich, St. Louis, MO, USA); BCA Protein Content Detection Kit (KGP902, KGI, Nanjing, China); anti-β-actin (20536-1-AP, Proteintech, Wuhan, China); anti-PREK (24390-1-AP, Proteintech, Wuhan, China); anti-GRP78 (11587-1-AP, Proteintech, Wuhan, China), HRP-labeled goat anti-rabbit IgG (SA00001-2, Proteintech, Wuhan, China); Fluorescein conjugated goat anti-rabbit IgG (SA00003-2, Proteintech, Wuhan, China); anti-IRE1α (3294S, Cell Signaling Technology, Danvers, MA, USA); anti-GSDMD-N (39754S, Cell Signaling Technology, Danvers, MA, USA); anti-Pro-Caspase1 (3866S, Cell Signaling Technology, Danvers, MA, USA); anti-ASC (13833S, Cell Signaling Technology, Danvers, MA, USA); anti-NLRP3 (15101S, Cell Signaling Technology, Danvers, MA, USA); anti-IL-1β (12703S, Cell Signaling Technology, Danvers, MA, USA); anti-Cleaved-IL-1β (83186S, Cell Signaling Technology, Danvers, MA, USA); anti-IL-18 (54943S, Cell Signaling Technology, Danvers, MA, USA); anti-ATF6 (DF6009, Affinity, Changzhou, China); anti-Cleaved Caspase1 (AF4005, Affinity, Changzhou, China); Normal Donkey Serum (SL050, Solarbio, Beijing, China); Hoechst 33,342 (C1028, Beyotime, Shanghai, China); Human IL-1 beta PicoKine ELISA Kit (EK0392, Boster, Wuhan, China); Human IL-18 PicoKine ELISA Kit (EK0864, Boster, Wuhan, China); CCK-8 Kit (abs50003, Absin, Shanghai, China); LDH Assay Kit (ab102526, Abcam, Cambridge, UK); TUDCA sodium (HY-19696A, MedChemExpress, Monmouth Junction, NJ, USA); MCC950 (HY-12815, MedChemExpress, Monmouth Junction, NJ, USA); Z-VAD-FMK (HY-16658B, MedChemExpress, Monmouth Junction, NJ, USA); VX-765 (S2228, Selleckchem, Houston, TX, USA).

### 4.2. BCG Culture

BCG was purchased from Chengdu Institute of Biological Products, China. To culture these bacteria, Middlebrook 7H9 medium containing 0.2% Tween-80, autoclaved and mixed with 10% ADC Enrichment medium, was used. BCG was then inoculated in this culture media and grown via static culture at 37 °C in a 5% CO_2_ cell incubator, with bacteria being harvested and aliquoted as reported in our prior manuscript [37].

### 4.3. Cell Culture and Infection

THP-1 cells were acquired from the Chinese Academy of Sciences Cell Bank and grown in RPMI-1640 medium with 10% FBS and 0.05 mmol·L^−1^ β-mercaptoethanol at 37 °C and 5% CO_2_ in a cell incubator, subcultured when the cell density reached 80–90%. To differentiate THP-1 monocytes into macrophages, the cells were cultured in 6-well plates (2 × 10^6^/well) containing PMA (50 ng/mL) for 48 h, after which they were cultured in fresh media for 24 h. Adherent cells were then infected with BCG (MOI = 10) following pretreatment for 2 h with appropriate inhibitors. After pretreatment with TUDCA (5 mM), MCC950 (10 μM), VX-765 (50 μM) or Z-VAD-FMK (50 μM) for 2 h, THP-1 macrophages were infected with BCG (MOI = 10) for subsequent testing.

### 4.4. Western Blotting

Prior to lysis, cells were washed thrice with chilled PBS. The resulting lysate was centrifuged at 12,000 rpm for 15 min. Protein concentration in the supernatant was determined using the BCA Protein Assay Kit. Subsequently, SDS-PAGE separation was performed and the separated proteins were transferred to activated PVDF membranes. Blots were blocked with 5% skimmed milk in TBS for 1 h at room temperature, followed by overnight incubation with antibodies specific for β-actin (1:3000), IRE1α (1:1000), PERK (1:1000), ATF6 (1:1000), GRP78(1:4000), GSDMD-N (1:1000), Pro-Caspase1 (1:1000), Cleaved Caspase1 (1:1000), NLRP3 (1:1000), ASC(1:1000), IL-1β (1:1000), Cleaved-IL-1β (1:1000) or IL-18 (1:1000). Blots were then rinsed 6 times with TBST (5 min/wash), probed for 1 h with HRP-labeled goat anti-rabbit IgG (1:3000) at room temperature and rinsed 4 times with TBST and 2 times with TBS (5 min/wash), then immunoreactive bands were visualized using an ECL kit from Abclonal (Wuhan, China). Protein bands were scanned with an Amersham Imager 6000 (GE Healthcare, Fairfield, CT, USA).

### 4.5. ELISA

The supernatants from cells in the appropriate treatment groups were collected and analyzed according to the instructions provided using ELISA kits.

### 4.6. CCK-8 Assay

Cells were plated in 96-well plates (1 × 10^4^/well). After treatment, as discussed above (See Section 2.3), 10 μL CCK-8 reagent was added per well. After a duration of 3 h, the absorbance was measured at 450 nm utilizing a fluorescent microplate reader, and the resulting data were recorded.

### 4.7. LDH Activity Assay

Cell culture supernatants were collected from appropriate treatment groups, and processed as per the kit instructions. Once completed, a microplate reader was used to measure the absorbance value (OD) at 450 nm every 5 min at 37 °C in the dark. The obtained data were recorded.

### 4.8. Immunofluorescent Detection of NLRP3

Cells were added to 12-well plates (3 × 10^5^/well) on coverslips. Appropriate inhibitors were used to pretreat these cells, after which they were infected using BCG (MOI = 10). At appropriate time points, the medium was removed and the cells were twice washed in PBS (3 min per wash), followed by fixation with paraformaldehyde (4%) for 20 min. Following three more PBS washes (3 min per wash), the cells were permeabilized in PBS containing Triton X-100 (0.5%) at room temperature for 20 min, washed again (3 × 3 min in PBS), blocked for 30 min with 10% Normal Donkey Serum, and probed overnight at 4 °C for NLRP3 expression. After that, the cells were rinsed with PBS (3 × 5 min), then probed with fluorescent secondary antibody at 37 °C for 1 h while being kept away from light, using Hoechst 33,342 as a nuclear counterstain, then washed with PBS (3 × 5 min) and observed under a microscope.

### 4.9. Transmission Electron Microscopy (TEM)

Cells were prefixed with a 3% glutaraldehyde, then postfixed in 1% osmium tetroxide, dehydrated in series acetone, infiltrated in Epox 812 for a longer time and embedded. The semithin sections were stained with methylene blue and ultrathin sections were cut with diamond knife, stained with uranyl acetate and lead citrate. Sections were examined with JEM-1400-FLASH transmission electron microscope.

### 4.10. Statistical Analysis

Experiments were completed in triplicate, and data were analyzed via t-tests or one-way ANOVAs using GraphPad Prism 9.0. The Tukey–Kramer multiple comparisons test was used for post hoc comparisons. * *p* < 0.05; ** *p* < 0.01; *** *p* < 0.001.

## 5. Conclusions

These results suggest that BCG infection can induce ERS activity within THP-1 macrophages, ERS regulates NLRP3 inflammasome activity and pyroptosis (Figure 7). Together, these results offer a robust foundation for future research aimed at exploring the etiology and treatment of TB.

## Figures and Tables

**Figure 1 ijms-24-11692-f001:**
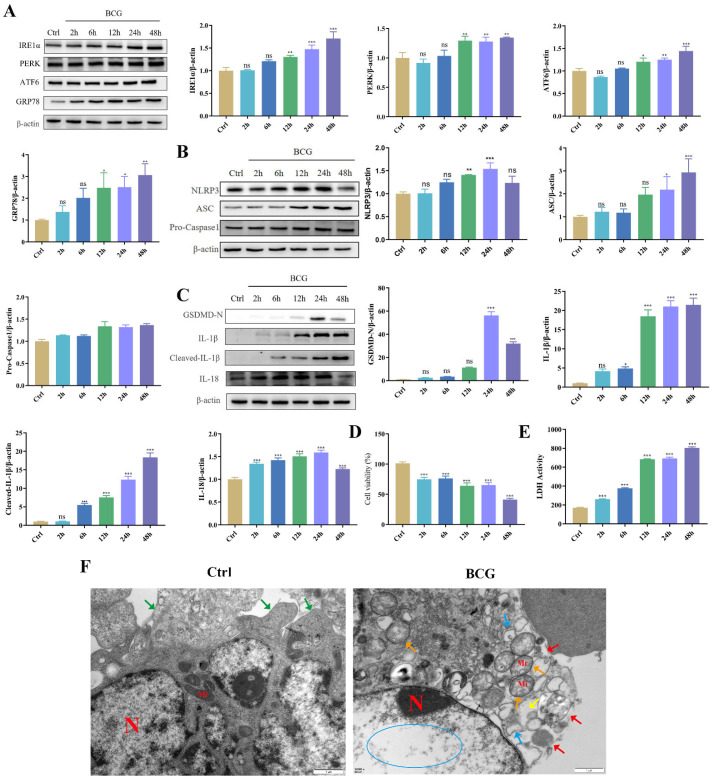
BCG infection induced THP-1 macrophage ERS, NLRP3 inflammasome activation and pyroptosis. (**A**) The expression of IRE1α, PERK, ATF6 and GRP78 in THP-1 macrophages was measured at indicated time points after BCG infection by Western blotting. (**B**) The expression of NLRP3, ASC and Pro-Caspase1 in THP-1 macrophages was measured at indicated time points after BCG infection by Western blotting (**C**) The expression of GSDMD-N, IL-1β, Cleaved-IL-1β and IL-18 in THP-1 macrophages was measured at indicated time points after BCG infection by Western blotting. (**D**) Cell viability at indicated time points after BCG infection of THP-1 macrophages was detected using CCK-8 kit. (**E**) LDH activity at indicated time points after BCG infection of THP-1 macrophages was detected using LDH kit. (**F**) Cell morphology in THP-1 macrophages infection with BCG for 24 h was observed of TEM. Scale bar: 1 μm. Nucleus (N); mitochondria (Mi); green arrows: cell membrane intact; red arrow: cell membrane rupture; yellow arrow: ribosome loss; orange arrow: mitochondrial swelling; blue arrow: rough endoplasmic reticulum expansion; blue circle: chromatin dissolution. The results of three replicate experiments are presented as means ± SEM. * *p* < 0.05; ** *p* < 0.01; *** *p* < 0.001.

**Figure 2 ijms-24-11692-f002:**
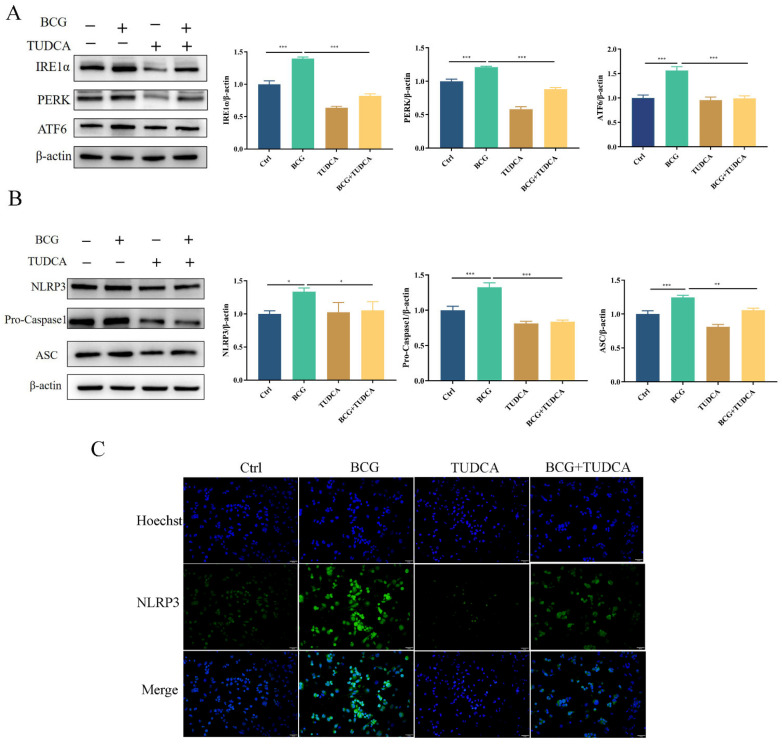
TUDCA treatment suppressed BCG-infected THP-1 macrophage ERS and NLRP3 inflammasome activation. THP-1 macrophages were pretreated for 2 h with TUDCA (5 mM) prior to BCG infection (MOI = 10) for 24 h (**A**). The expression of IRE1α, PERK and ATF6 was measured by Western blotting (**B**) The expression of NLRP3, Pro-Caspase1 and ASC was measured by western blotting. (**C**) Immunofluorescent staining was utilized to detect NLRP3 in these cells. Scale bar: 50 μm. Data are means ± SEM from triplicate experiments. * *p* < 0.05; ** *p* < 0.01; *** *p* < 0.001.

**Figure 3 ijms-24-11692-f003:**
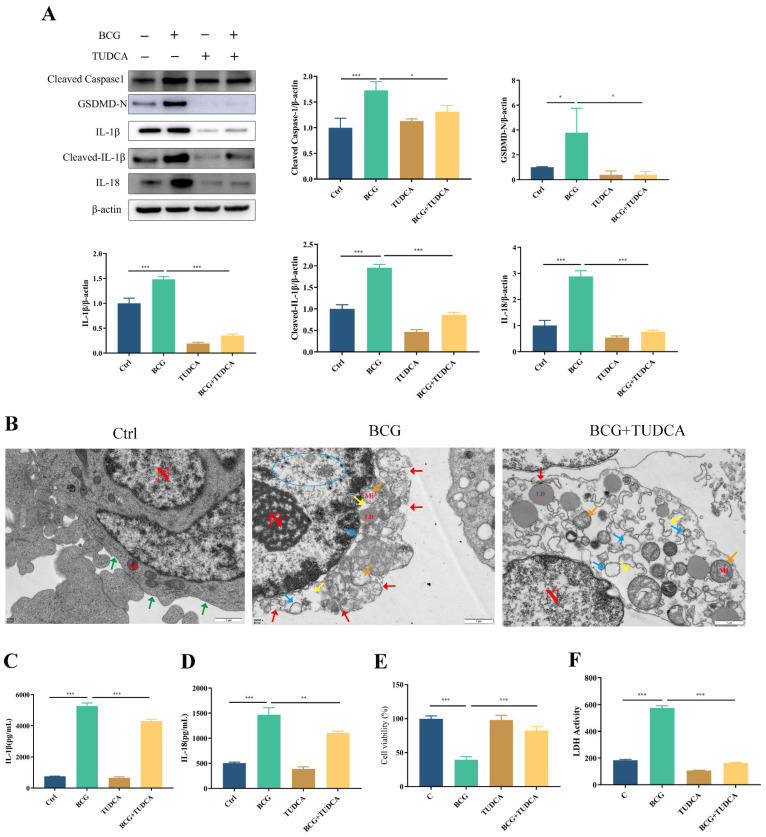
TUDCA treatment inhibited BCG-infected THP-1 macrophage pyroptosis. THP-1 macrophages were pretreated for 2 h with TUDCA (5 mM) prior to BCG infection (MOI = 10) for 24 h. (**A**) The expression of Cleaved Caspase1, GSDMD-N, IL-1β, Cleaved-IL-1β and IL-18 was assessed by Western blotting. (**B**) TEM observation of cell morphology. Scale bar: 1 μm. Nucleus (N); mitochondria (Mi); green arrows: cell membrane intact; red arrow: cell membrane rupture; yellow arrow: ribosome loss; orange arrow: mitochondrial swelling; blue arrow: rough endoplasmic reticulum expansion; blue circle: chromatin dissolution. (**C**,**D**) Cell culture supernatant IL-1β and IL-18 concentrations were detected by ELISA. (**E**) Cell viability in THP-1 macrophages was detected using CCK-8 kit. (**F**) LDH activity in THP-1 macrophages was detected using LDH kit. Data are means ± SEM from triplicate experiments. * *p* < 0.05; ** *p* < 0.01; *** *p* < 0.001.

**Figure 4 ijms-24-11692-f004:**
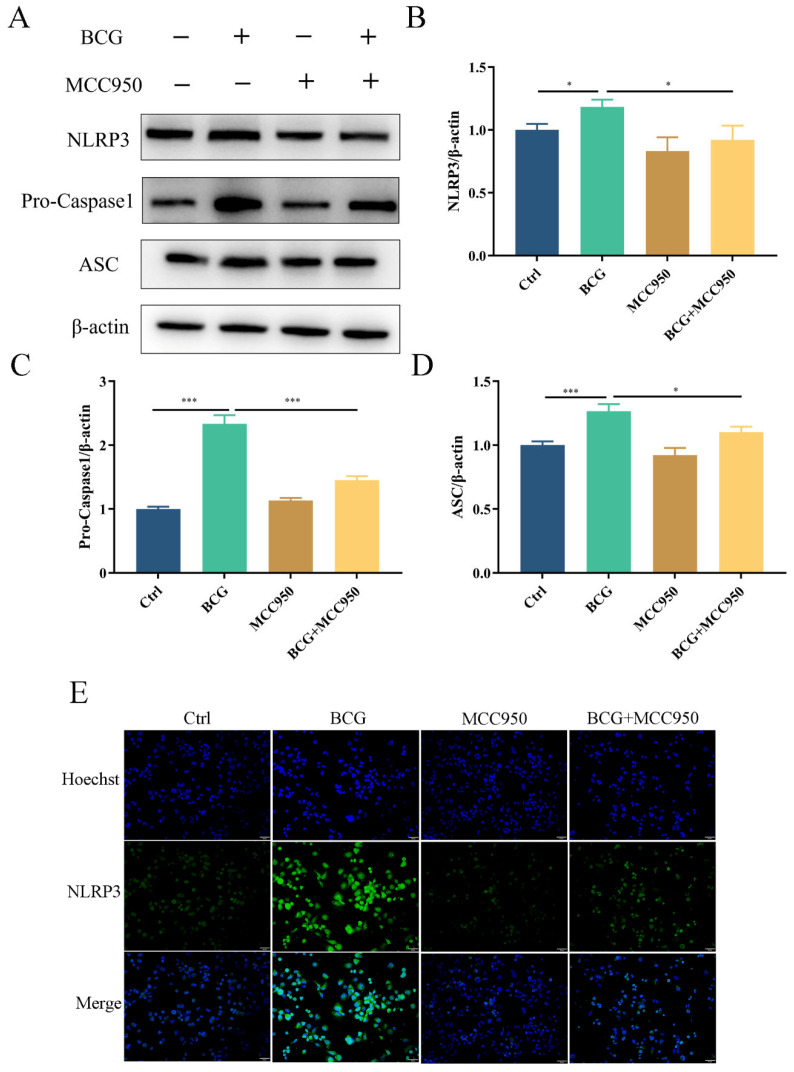
MCC950 inhibited BCG-infected THP-1 macrophages NLRP3 inflammasome activation. (**A**–**D**) After pretreatment for 2 h with MCC950 (10 μM), THP-1 macrophages were infected for 24 h with BCG (MOI = 10), after which NLRP3, Pro-Caspase1 and ASC protein levels were examined by Western blotting. (**E**) Immunofluorescent staining was utilized to detect NLRP3 in these cells. Scale bar: 50 μm. The data are presented as means ± SEM from triplicate experiments. * *p* < 0.05; *** *p* < 0.001.

**Figure 5 ijms-24-11692-f005:**
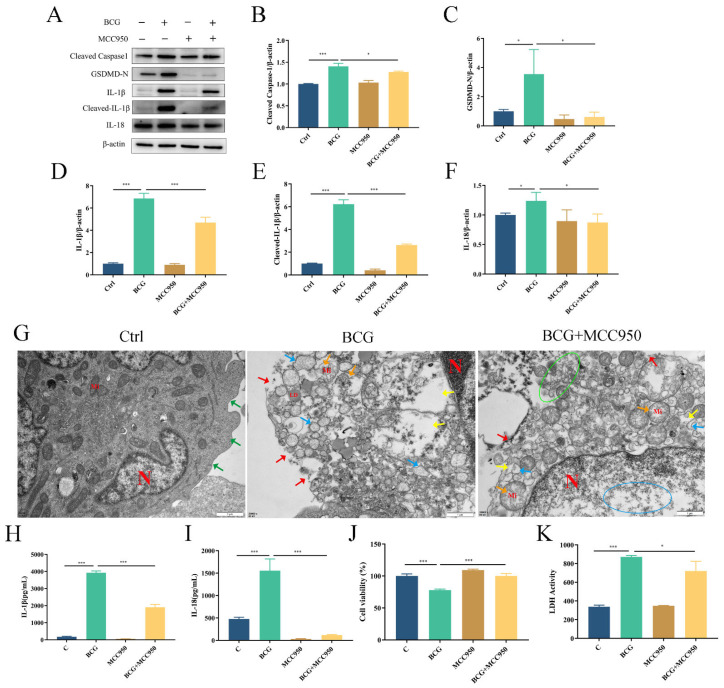
NLRP3 inflammasome-mediated pyroptosis in BCG-infected THP-1 macrophages. THP-1 macrophages were pretreated with MCC950 (10 μM) for 2 h and infected with BCG (MOI = 10) for 24 h. (**A**–**F**) The expression of Cleaved Caspase1, GSDMD-N, IL-1β, Cleaved-IL-1β and IL-18 protein levels were assessed by Western blotting (**G**) TEM observation of cell morphology. Scale bar: 1 μm. nucleus(N); mitochondria (Mi); green arrows: cell membrane intact; red arrow: cell membrane rupture; yellow arrow: ribosome loss; orange arrow: mitochondrial swelling; blue arrow: rough endoplasmic reticulum expansion; green circle: extravasation of cytoplasmic contents. (**H**,**I**) Cell culture supernatant IL-1β and IL-18 concentrations were detected by ELISA. (**J**) THP-1 macrophage viability was detected using CCK-8 kit. (**K**) LDH activity in THP-1 macrophages was detected using LDH kit. Data are means (±SEM) of three independent experiments. * *p* < 0.05; *** *p* < 0.001.

**Figure 6 ijms-24-11692-f006:**
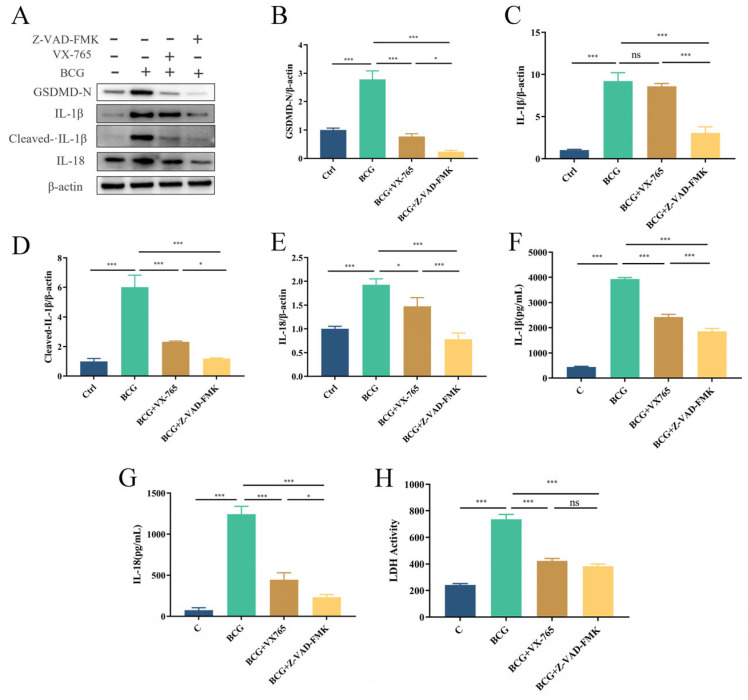
BCG infection promotes THP-1 macrophages pyroptosis via both canonical and noncanonical pathways. Cells were pretreated with 50 μM of VX-765 or 50 μM of Z-VAD-FMK for 2 h; they were then infected with BCG (MOI = 10) for 24 h. (**A**–**E**) The expression of GSDMD-N, IL-1β, Cleaved-IL-1β and IL-18 was measured by Western blotting. (**F**,**G**) Cell culture supernatant IL-1β and IL-18 concentrations were detected by ELISA. (**H**) LDH activity in THP-1 macrophages was detected using LDH kit. The data are presented as means ± SEM from triplicate experiments. * *p* < 0.05; *** *p* < 0.001.

**Figure 7 ijms-24-11692-f007:**
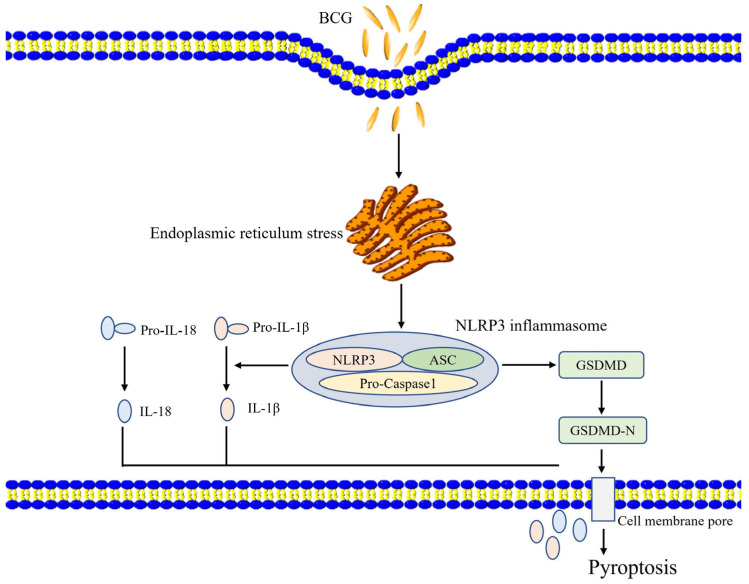
A proposed model of the ERS’s role in BCG-infected cells. BCG infection stimulates ERS signaling in THP-1 macrophages, thereby activating the NLRP3 inflammasome and inducing pyroptotic death via canonical and noncanonical pathways.

## Data Availability

The original contributions presented in the study are included in the article.

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
