# Peer review of "Endoplasmic Reticulum Stress Mediated NLRP3 Inflammasome Activation and Pyroptosis in THP-1 Macrophages Infected with *Bacillus Calmette-Guérin"

_ijms, 2023, doi:10.3390/ijms241411692_

Round 1
Reviewer 1 Report
This study investigated the mechanisms of pyroptosis by BCG infection in THP-1 cells. The authors performed experiments thoroughly. However, there were still some comments to the authors.
1. How did the authors tell the border of cell membrane and the release of content by TEM pictures in Figure 1F? The authors should label the nucleus and dress more about the results. In addition, why the yellow arrows pointed the empty area in
2. TUDCA alone treatment strongly reduced IRE1a and PERK in Fig. 2A. However, this phenomenon was not observed in other proteins, including NLRP3 and ASC. Could the authors explain it?
3. The fonts in all figures should be uniform.
4. Since the authors drew the release of IL-18 and IL-1b in the Fig. 7, the release of IL-18 and IL-1b should be detected by ELISA.
Author Response
Response to Reviewer 1 Comments
we would like to thank you for your letter on our article (Manuscript ID: ijms-2461929).

Reviewer 2 Report
The manuscript entitled “Endoplasmic reticulum stress mediated NLRP3 inflammasome activation and pyroptosis in THP-1 macrophages infected with Bacillus Calmette-Guérin” evaluated the activation of ERS, NLRP3 inflammasome and pyroptosis in BCG-infected macrophages. Although the study has an adequate methodology for the proposed objectives and has results that endorse the current literature, the manuscript lacks new information in relation to previously published studies. (Liao et al. Endoplasmic Reticulum Stress Induces Macrophages to Produce IL-1β During Mycobacterium bovis Infection via a Positive Feedback Loop Between Mitochondrial Damage and Inflammasome Activation. Front Immunol. 2019; Li et al. Tanshinone IIA alleviates NLRP3 inflammasome-mediated pyroptosis in Mycobacterium tuberculosis-(H37Ra-) infected macrophages by inhibiting endoplasmic reticulum stress. J Ethnopharmacol.2022; Fu et al. Inhibition of the PERK/TXNIP/NLRP3 Axis by Baicalin Reduces NLRP3 Inflammasome-Mediated Pyroptosis in Macrophages Infected with Mycobacterium tuberculosis. Mediators Inflamm. 2021).
Main considerations:
- Revise the English of the manuscript.
Abstract:
Line 17: Meaning of ERS.
Introduction
Line 44: Meaning of ERS.
Line 45: Remove “endoplasmic reticulum”
Line 46: Remove “endoplasmic reticulum stress”.
Lines 62-63: remove “endoplasmic reticulum stress” and “unfolded protein response”.
Line 73: Meaning of BCG.
The authors do not contextualize the use of BCG in the study and only mention its use as the objective of the work in the last paragraph. The introduction needs to better demonstrate the originality of the work and the use of BCG as a study tool for Mtb infection.
Material and methods
Lines 256-257: Enter complete reagent information, including catalog number for antibodies.
Line 324: Correct "tissue" to "cells"
Line 331: What statistical test was used after performing ANOVA?
Results
Lines 84-85; 111: The authors evaluated only the expression of the transmembrane proteins IRE1α, PERK and ATF6, which indicate only the initial activation of the UPR. As performed in other studies, it is necessary to evaluate the expression of GRP78 and especially CHOP to suggest ERS activation, since CHOP is involved in the final ERS pathway.
Figure 1: Authors need to enlarge TEM images to allow better visualization of ultramicroscopy.
Line 154: Correct “TUDCA treatment” by “MCC950 treatment”.
Discussion:
Line 216: Correct to “Caspase 4/5/11”.
The authors only cite the results at the end of each paragraph and do not adequately discuss them according to the studies already published.
Revise the English of the manuscript.
Author Response
I have already uploaded.

Reviewer 3 Report
This manuscript by Nie X et al. presents findings on the role of BCG infection in THP1 macrophages in mediating ER stress and its subsequent impact on NLRP3 inflammasome activation and proptosis.
Previously, there have been indications that Mtb infection triggers inflammasome activation and pyroptosis, as evidenced by several studies available on PubMed (PMID: 22101787, PMID: 34324582). Some of these studies have also discussed how Mtb infection manipulates the host cell inflammasome, mainly through the inhibition of AIM2 and NLRP3 inflammasomes. It is worth noting that previous research has suggested that BCG cannot activate the inflammasome (PMID: 22101787), and it would be valuable for the authors to comment on this discrepancy.
Overall, the novelty of the findings reported in this manuscript may be debatable.
The most intriguing aspect lies in the observation of ER stress and its potential role in mediating inflammasome activation following BCG infection.
In Figure 3C and D, even after TUDCA incubation, the levels of IL-1β and IL-18 remain significantly induced, with only slight effects on viability and no impact on LDH release. The authors should provide an explanation for this finding. Additionally, it would be interesting to know if ER stress affects ROS production in this context.
Furthermore, the authors should explore the potential effects of BCG on the NFKB pathway and TLR4 activation. This information would be valuable in understanding the broader impact of BCG infection on macrophage signaling.
The evidence presented in the manuscript strongly supports the involvement of non-canonical inflammasome activation, as indicated by the VX765 data. However, the discussion of this aspect is lacking and could be improved by providing a more comprehensive analysis and critical assessment of previous findings in the field.
To enhance the discussion, it would be beneficial for the authors to delve into the existing literature and contextualize their findings. By doing so, the authors can highlight the novelty and significance of their specific results while acknowledging the existing knowledge in the field.
Additionally, the introduction of pyroptosis and the NLRP3 inflammasome in the discussion section appears to be redundant (lines 213-220 and again 226), and these could be better placed in the introduction section.
Some typos and repetitions are present in the manuscript and need to be corrected.
Author Response
Response to Reviewer 3 Comments
First, we would like to thank you for your letter and constructive comments on our article (Manuscript ID: ijms-2461929). These comments are all valuable for further improving our article. We have carefully discussed these comments and made revisions one by one according to the comments. In order to facilitate your review, the revised sections have been highlighted with yellow and the updated version has been submitted to the journal's submission system. Detailed revision instructions and answers are as follows:
Comments and Suggestions for Authors
Point 1: Previously, there have been indications that Mtb infection triggers inflammasome activation and pyroptosis, as evidenced by several studies available on PubMed (PMID: 22101787IF: 6.688 Q2 , PMID: 34324582IF: 7.464 Q1 ). Some of these studies have also discussed how Mtb infection manipulates the host cell inflammasome, mainly through the inhibition of AIM2 and NLRP3 inflammasomes. It is worth noting that previous research has suggested that BCG cannot activate the inflammasome (PMID: 22101787IF: 6.688 Q2 ), and it would be valuable for the authors to comment on this discrepancy.
Overall, the novelty of the findings reported in this manuscript may be debatable.
Response 1: Thank you for your valuable comments. We have carefully read the article you recommended (PMID:22101787IF:6.688 Q2). The results of the article show that the process of tubercle bacilli specifically activation of NLRP3 inflammasome is strictly controlled by the virulence-associated RD1 locus of MTB. Due to the deletion of RD1 region in BCG, BCG cannot activate inflammasome after infection of BMDMs. Other studies showed that, compared with the control group, the expression of NLRP3 inflammasome-related indicators was up-regulated after BCG-infected macrophages[1-3]. Therefore, we consider that the activation of inflammasome by BCG-infected macrophages is a problem worthy of further exploration. This may be related to the missing RD1 fragment of BCG, and may also be related to the multiplicity of infection, infection time, infection model, cellular inflammatory microenvironment and BCG itself.
[1] Qu Z, Zhou J, Zhou Y, Xie Y, Jiang Y, Wu J, Luo Z, Liu G, Yin L, Zhang XL. Mycobacterial EST12 activates a RACK1-NLRP3-gasdermin D pyroptosis-IL-1β immune pathway. Sci Adv. 2020 Oct 23;6(43):eaba4733. doi: 10.1126/sciadv.aba4733. PMID: 33097533; PMCID: PMC7608829.
[2] Souza De Lima D, Bomfim CCB, Leal VNC, Reis EC, Soares JLS, Fernandes FP, Amaral EP, Loures FV, Ogusku MM, Lima MRD, Sadahiro A, Pontillo A. Combining Host Genetics and Functional Analysis to Depict Inflammasome Contribution in Tuberculosis Susceptibility and Outcome in Endemic Areas. Front Immunol. 2020 Oct 21;11:550624. doi: 10.3389/fimmu.2020.550624. PMID: 33193317; PMCID: PMC7609898.
[3] Pu W, Zhao C, Wazir J, Su Z, Niu M, Song S, Wei L, Li L, Zhang X, Shi X, Wang H. Comparative transcriptomic analysis of THP-1-derived macrophages infected with Mycobacterium tuberculosis H37Rv, H37Ra and BCG. J Cell Mol Med. 2021 Nov;25(22):10504-10520. doi: 10.1111/jcmm.16980. Epub 2021 Oct 10. PMID: 34632719; PMCID: PMC8581329.
Point 2: The most intriguing aspect lies in the observation of ER stress and its potential role in mediating inflammasome activation following BCG infection.
In Figure 3C and D, even after TUDCA incubation, the levels of IL-1β and IL-18 remain significantly induced, with only slight effects on viability and no impact on LDH release. The authors should provide an explanation for this finding. Additionally, it would be interesting to know if ER stress affects ROS production in this context.
Furthermore, the authors should explore the potential effects of BCG on the NFKB pathway and TLR4 activation. This information would be valuable in understanding the broader impact of BCG infection on macrophage signaling.
Response 2: Thank you for your valuable comments. Regarding the question you raised, we consider the possible reasons are as follows: during the process of pyroptosis induced by BCG-infected THP-1 macrophages, the expression of inflammatory factors IL-1β and IL-18 may be induced through multiple pathways, one of which involves ERS. Therefore, when cells were pretreated with TUDCA, a specific inhibitor of ERS, it has a limited inhibitory effect on the expression of inflammatory factors caused by BCG infection. In assessing various modes of cell death (including apoptosis, pyroptosis, necrosis, etc.), CCK-8 and LDH are usually used for detection. Studies have shown that TUDCA can increase cell survival by reducing apoptosis[1]. Our results also showed that TUDCA could inhibit pyroptosis induced by BCG infection. Therefore, we speculate that in the process of BCG infection of macrophages, in addition to mediating pyroptosis, ERS may also regulate other types of cell death. This hypothesis needs to be confirmed by our follow-up in-depth research.
[1] Zhang W, Chen L, Feng H, Wang W, Cai Y, Qi F, Tao X, Liu J, Shen Y, Ren X, Chen X, Xu J, Shen Y. Rifampicin-induced injury in HepG2 cells is alleviated by TUDCA via increasing bile acid transporters expression and enhancing the Nrf2-mediated adaptive response. Free Radic Biol Med. 2017 Nov;112:24-35. doi: 10.1016/j.freeradbiomed.2017.07.003. Epub 2017 Jul 6. PMID: 28688954.
We are very grateful for the research ideas you provided us. Previous studies in our laboratory showed that BCG infection of RAW264.7 cells induced ROS production, but did not explore whether ERS affects ROS production[1]. Studies have shown that the occurrence of ERS can induce the production of ROS[2,3]. Therefore, we will further explore whether ERS induces ROS generation after BCG infection of THP-1 macrophages. Regarding the effect of BCG on the NF-κB pathway and TLR4 activation after infection of THP-1 macrophages, the laboratory has explored that BCG has a certain regulatory effect on the NF-κB pathway after infection of THP-1 macrophages[4]. However, the interaction between NF-κB signaling pathway and TLR4 was not explored. Next, we will further explore the interaction between NF-κB pathway and TLR4 after BCG-infected THP-1 macrophages as another new project according to your opinions.
[1] Yu J, Ma C, Xu Y, Han L, Wu X, Wang Y, Deng G. Knockdown of fatty acid binding protein 4 exacerbates Bacillus Calmette-Guerin infection-induced RAW264.7 cell apoptosis via the endoplasmic reticulum stress pathway. Infect Genet Evol. 2020 Nov;85:104552. doi: 10.1016/j.meegid.2020.104552. Epub 2020 Sep 11. PMID: 32920196.
[2] Ming S, Tian J, Ma K, Pei C, Li L, Wang Z, Fang Z, Liu M, Dong H, Li W, Zeng J, Peng Y, Gao X. Oxalate-induced apoptosis through ERS-ROS-NF-κB signalling pathway in renal tubular epithelial cell. Mol Med. 2022 Aug 3;28(1):88. doi: 10.1186/s10020-022-00494-5. PMID: 35922749; PMCID: PMC9347104.
[3] Liao Y, Hussain T, Liu C, Cui Y, Wang J, Yao J, Chen H, Song Y, Sabir N, Hussain M, Zhao D, Zhou X. Endoplasmic Reticulum Stress Induces Macrophages to Produce IL-1β During Mycobacterium bovis Infection via a Positive Feedback Loop Between Mitochondrial Damage and Inflammasome Activation. Front Immunol. 2019 Feb 21;10:268. doi: 10.3389/fimmu.2019.00268IF: 8.786 Q1 . PMID: 30846986; PMCID: PMC6394253.
[4] Liu Z, Wang J, Dai F, Zhang D, Li W. DUSP1 mediates BCG induced apoptosis and inflammatory response in THP-1 cells via MAPKs/NF-κB signaling pathway. Sci Rep. 2023 Feb 14;13(1):2606. doi: 10.1038/s41598-023-29900-6. PMID: 36788275; PMCID: PMC9926451.
Point 3: The evidence presented in the manuscript strongly supports the involvement of non-canonical inflammasome activation, as indicated by the VX765 data. However, the discussion of this aspect is lacking and could be improved by providing a more comprehensive analysis and critical assessment of previous findings in the field.
Response 3: Thank you for your comments. Based on your comments, we have conducted a more comprehensive analysis of this part in the discussion. Add “Previous studies have demonstrated that canonical and non-canonical pathways of pyroptosis are associated with various diseases. including I/R injury and Periodontitis. Therefore, we explored the pathway of pyroptosis after BCG-infected THP-1 macrophages. In this study, we pretreated cells with the VX-765 and Z-VAD-FMK, to investigate their effects on pyroptosis. Our results showed that both inhibitors effectively inhibited pyroptosis. However, Z-VAD-FMK exhibited a greater down-regulation of pyroptosis-related proteins compared to VX-765. These findings suggest that the pathways of pyroptosis induced by BCG infection of THP-1 macrophages involve both canonical and non-canonical pathways” to end of Discussion second paragraph.
Point 4:To enhance the discussion, it would be beneficial for the authors to delve into the existing literature and contextualize their findings. By doing so, the authors can highlight the novelty and significance of their specific results while acknowledging the existing knowledge in the field.
Additionally, the introduction of pyroptosis and the NLRP3 inflammasome in the discussion section appears to be redundant (lines 213-220 and again 226), and these could be better placed in the introduction section.
Response 4: Thank you for your comments. According to your suggestion, we have moved the introduction of pyroptosis from the discussion to the introduction, and further improved the writing of the discussion. Add“This process is marked by the creation of pores in the cell membrane, the expulsion of cellular contents, and the release of IL-1β and IL-18. Moreover, Caspase3/8 can induce pyroptosis, with the former cleaving GSDME to generate the pyroptotic mediator GSDME-N, and the latter cleaving GSDMD in response to TAK1 inhibition” before line 77 “Many factors have been shown to stimulate pyroptosis”.
Round 2
Reviewer 1 Report
None.
Author Response

(The authors gave the same response as above.)

Reviewer 2 Report
The authors adequately responded to the suggestions. However, I suggest including GRP78 and CHOP data in the manuscript or properly referencing the work already carried out by the group so that they can suggest the occurrence of ERS.
The English of the manuscript needs minor adjustments.
Author Response
Response to Reviewer 2 Comments
First, we would like to thank you for your letter and constructive comments on our article (Manuscript ID: ijms-2461929). These comments are all valuable for further improving our article. We have carefully discussed these comments and made revisions one by one according to the comments. In order to facilitate your review, the revised sections have been highlighted with red and the updated version has been submitted to the journal's submission system. Detailed revision instructions and answers are as follows:
Point 1: The authors adequately responded to the suggestions. However, I suggest including GRP78 and CHOP data in the manuscript or properly referencing the work already carried out by the group so that they can suggest the occurrence of ERS.
Response 1: Thanks for your suggestion. The result for GRP78 have been added in Fig 1A. Regarding the result of CHOP, an article published by our laboratory has been cited in the article (Reference 24). Bold font indicates me.
Ma, B.; Liu, Y.; Nie, X.; Li, M.; Yang, Y.; Xu, J. Regulation of ATF6/CHOP pathway on pyroptosis in THP-1 cells infected with Mycobacterium Bacillus Calmette-Guérin. Chinese journal of pathopmysiology. 2022, 38, 2183-2190
Point 2: The English of the manuscript needs minor adjustments.
Response 2: Thanks for your suggestion. We have carefully read the full text and made adjustments to the English language of the manuscript.
Reviewer 3 Report
Please remove lines 265-266 as it is established in the field that NLRP3 inflammasome activation mediates pyroptosis. Add a citation of this instead i.e. PMID: 31284572.
Author Response
Response to Reviewer 3 Comments
First, we would like to thank you for your letter and constructive comment on our article (Manuscript ID: ijms-2461929). Your comment is valuable for further improving our article. We have made changes based on your comment. For your review, the revised parts have been highlighted in red, and the updated version has been submitted to the journal submission system. Detailed revision instructions and answers are as follows:
Point: Please remove lines 265-266 as it is established in the field that NLRP3 inflammasome activation mediates pyroptosis. Add a citation of this instead i.e. PMID: 31284572.
Response: Thank you for your suggestion. Based on your suggestion, we have removed lines 265-266 and added a reference to PMID: 31284572.